# Effectiveness of a Novel Continuous Doppler (Moyo) Versus Intermittent Doppler in Intrapartum Detection of Abnormal Foetal Heart Rate: A Randomised Controlled Study in Tanzania

**DOI:** 10.3390/ijerph16030315

**Published:** 2019-01-24

**Authors:** Benjamin Kamala, Hussein Kidanto, Ingvild Dalen, Matilda Ngarina, Muzdalifat Abeid, Jeffrey Perlman, Hege Ersdal

**Affiliations:** 1Faculty of Health Sciences, University of Stavanger, 4036 Stavanger, Norway; hege.ersdal@safer.net; 2Muhimbili National Hospital, P.O. Box 65000, Dar es Salaam, Tanzania; mmatty71@gmail.com; 3Department of Research, Stavanger University Hospital, 4011 Stavanger, Norway; hkidanto@gmail.com (H.K.); ingvild.dalen@sus.no (I.D.); 4School of Medicine, Aga Khan University, P.O. Box 38129, Dar es Salaam, Tanzania; amuzdalifat29@gmail.com; 5Department of Paediatrics, Weill Cornell Medicine, New York, NY 10065, USA; jmp2007@med.cornell.edu; 6Department of Anaesthesiology and Intensive Care, Stavanger University Hospital, 4011 Stavanger, Norway

**Keywords:** foetal heart rate, Moyo, Doppler, perinatal outcomes

## Abstract

*Background*: Intrapartum foetal heart rate (FHR) monitoring is crucial for identification of hypoxic foetuses and subsequent interventions. We compared continuous monitoring using a novel nine-crystal FHR monitor (Moyo) versus intermittent single crystal Doppler (Doppler) for the detection of abnormal FHR. *Methods*: An unmasked randomised controlled study was conducted in a tertiary hospital in Tanzania (ClinicalTrials.gov Identifier: NCT02790554). A total of 2973 low-risk singleton pregnant women in the first stage of labour admitted with normal FHR were randomised to either Moyo (*n* = 1479) or Doppler (1494) arms. The primary outcome was the proportion of abnormal FHR detection. Secondary outcomes were time intervals in labour, delivery mode, Apgar scores, and perinatal outcomes. *Results*: Moyo detected abnormal FHR more often (13.3%) compared to Doppler (9.8%) (*p* = 0.002). Time intervals from admission to detection of abnormal FHR were 15% shorter in Moyo (*p* = 0.12) and from the detection of abnormal FHR to delivery was 36% longer in Moyo (*p* = 0.007) compared to the Doppler arm. Time from last FHR to delivery was 12% shorter with Moyo (*p* = 0.006) compared to Doppler. Caesarean section rates were higher with the Moyo device compared to Doppler (*p* = 0.001). Low Apgar scores (<7) at the 1st and 5th min were comparable between groups (*p* = 0.555 and *p* = 0.800). Perinatal outcomes (fresh stillbirths and 24-h neonatal deaths) were comparable at delivery (*p* = 0.497) and 24-h post-delivery (*p* = 0.345). *Conclusions*: Abnormal FHR detection rates were higher with Moyo compared to Doppler. Moyo detected abnormal FHR earlier than Doppler, but time from detection to delivery was longer. Studies powered to detect differences in perinatal outcomes with timely responses are recommended.

## 1. Introduction

The intrapartum period poses a great risk for both baby and mother. Globally, 2.6 million neonates die annually during the neonatal period, accounting for approximately 46% of all under-five deaths [1]. Approximately 36% of these neonates die on the first day [1,2], and 25% are intrapartum related [1,3]. Moreover, 40% of 2.6 million stillbirths are intrapartum related and termed fresh stillbirths (FSB) [4]. Most of these perinatal deaths occur in low income countries (LIC) where effective emergency obstetric care provision is low [3].

Prolonged intrapartum foetal hypoxia, invariably because of interruption of placental blood flow, may result in a FSB or a severely asphyxiated neonate [5,6]. Following delivery, such neonates may die, or survive with variable degrees of hypoxic-ischemic brain injury, leading to long-term neurocognitive and behavioural impairment [7,8].

Intrapartum foetal heart rate (FHR) monitoring is an important strategy in providing a more targeted and appropriate management of foetal well-being [9]. Intermittent assessment with either a hand-held Doppler or Pinard Stethoscope is the main method of intrapartum FHR monitoring in LIC [10,11]. However, studies document that intrapartum FHR monitoring is not conducted according to current international guidelines [12,13,14], due to a shortage of human resource and appropriate monitoring equipment [12,15], leading to perinatal morbidity and mortality [16].

Our previous studies using a novel continuous Doppler (Moyo) showed that accurate FHR monitoring enhanced early detection of the at-risk foetus [17,18]. The Moyo device has features that may facilitate early identification of foetuses at risk of intra-partum hypoxia and improve the quality of midwifery practices [17]. Continuous intrapartum FHR monitoring with cardiotocograph (CTG) coupled with timely interventions, such as caesarean sections in high income countries, has been associated with improved perinatal outcomes [9]. There is a paucity of studies on FHR monitoring in LIC, where most births occur, posing a critical need to implement and test new monitoring strategies in these settings [19]. An ideal device for LIC settings should be low-cost, simple to operate, possible to operate on a range of power sources, and without a need for continuous power supply, which the Moyo device represents [20].

We hypothesized that continuous intrapartum monitoring with Moyo as compared to intermittent Doppler assessment would lead to more timely and frequent detection of FHR abnormalities. The objective of the study was to compare the effectiveness of continuous monitoring (Moyo-intervention) versus intermittent hand-held (Doppler-standard of care) in the intrapartum detection of abnormal FHR.

## 2. Materials and Methods

### 2.1. Study Design

We conducted a parallel-arms, unmasked randomised controlled study from March 2016 to September 2017 at Muhimbili National Hospital (MNH) in Dar es Salaam, Tanzania.

### 2.2. Study Settings

MNH is the national referral hospital and a teaching hospital for Muhimbili University. The hospital provides both basic and comprehensive emergency obstetric care and has approximately 10,000 annual deliveries; 50% by Caesarean section (CS) [10]. The labour ward at MNH has 20 delivery beds and approximately 25 nurse midwives. The ward is managed by 5 nurse-midwives and 2 nursing assistants in each shift of 12 h. The doctors-on-call team comprises 1 consultant, 1 obstetrician, 2 obstetric residents, and 1 intern doctor on 24-h call. There are two obstetric operating theatres in a separate building adjacent to the maternity block.

On admission, a nurse midwife screens all women for vital signs registration, initial FHR assessment, and vaginal examination before entering the labour ward. A brief history and vital signs are taken and required information entered in the labour ward register. The on-call doctor reviews the partograph and undertakes the initial and subsequent obstetric examination until delivery. After a normal vaginal delivery, mothers and babies are observed in the hospital for 6–10 h. Babies with respiratory distress and others in need of medical attention are admitted to the neonatal unit. Management protocols for mothers and babies in this setting have been described previously [21].

### 2.3. Study Participants

The study participants included mothers in labour with an estimated gestational age above 28 weeks and with ≥3 cm cervical dilatation. Exclusions included scheduled elective CS, multiple pregnancies, cases with abnormal or undetectable FHR on admission, admission in the second stage with full cervical dilation, precipitous delivery, and critically ill patients with no measurements of FHR.

### 2.4. Patient and Public Involvement

The need for development of the Moyo device started at Haydom, a rural based hospital in Northern Tanzania, and MNH responding to increased intrapartum related perinatal morbidity and mortality [5,6,22]. The device was developed in collaboration with clinical staff at these hospitals, Laerdal Medical, and Stavanger University Hospital in Norway. It was in response to the needs of the clinical staff and mothers in these resource limited settings to reduce FSB and END (early neonatal deaths). Patients were told of the design of the study before being recruited to participate. Qualitative studies on preferences and acceptability of the continuous FHR monitoring with the Moyo device among mothers and clinical staff have been conducted in these settings. Positive responses on this device compared to the traditional Pinard stethoscope and Doppler were obtained and documented in our previous studies [23,24].

### 2.5. Randomisation, Concealment, and Masking

A randomisation sequence was computer-generated by an independent statistician. Details of the allocated group were given to the study coordinator, who supervised data clerks to write on cards and put them in sequentially numbered opaque sealed envelopes and sealed them. The allocation sequence was concealed from investigators and nurses enrolling participants and assessing outcomes. Envelopes were prepared and stored in a locked cabinet. Consecutively numbered envelopes were opened only after the enrolled participants completed assessments. The women and enrolling nurses were unaware of the allocation group until after eligible women were informed about the study and a written consent was obtained. Women, nurses, and doctors were unmasked.

### 2.6. Training

Pre-study trainings using a Moyo training package focusing on standard operating procedures for Moyo and international FHR monitoring standards were conducted in January and February 2016 by study investigators. All labour ward staff were trained for a full day on these FHR management protocols before starting the study. Continuous on-job refresher trainings were conducted (every two months) to increase protocol adherence and accommodate incoming staff who did not receive the initial training. Training included theoretical information about FHR monitoring during labour and management of an abnormal FHR. Criteria for FHR monitoring were established and included monitoring recording every 30 min in the first stage of labour, and every 5–15 min in the second stage [12,13,14]. The labour ward staff were also told that abnormal FHR detections should be reported to the doctor on call, who should act according to hospital protocols. Research nurses (at least 2 per shift) were trained for one additional day on research protocol and data collection to ensure accuracy and completeness of the data in the paper-based case report form (CRF). Data were collected from mothers’ antenatal cards, partograph, obstetric register, and, when needed, from routine neonatal morbidity and mortality records in the neonatal unit.

### 2.7. The Intervention (The Moyo Device)

Moyo (Figure 1 and Figure 2) (Moyo, Laerdal Global Health, Stavanger, Norway) is a novel strap-on FHR monitor equipped with a rechargeable battery, containing a nine-crystal Doppler ultrasound sensor, which facilitates the rapid identification of FHR within 5 s. Additional features of the Moyo device have been described in our previous studies [17,18,23].

Women randomised to the Moyo arm received information on how the device was to be used by the enrolling midwife before the device was strapped on. The midwife continued with her routine activities, but periodically revisited the women to check and record the FHR reading or in case of an abnormal FHR alarm from Moyo [23]. Moyo continued to be strapped on until the end of the second stage or immediately prior to the start of a CS.

### 2.8. Control (Hand-Held Doppler)

In the control arm, women were monitored intermittently with the standard protocol of FHR monitoring every 30 min in the first stage and 5–15 min in the second stage using a hand-held Doppler (Power-free Education Technology, Pet.og.za, Cape Town, South Africa). Doppler detects FHR and provides a steady state number per min on a display, as well as an audible sound of the FHR [11]. It permits the midwife to locate the FHR while allowing others, including the mother, to hear the FHR. The midwife would continue with her routine activities and periodically revisit the women to check and record FHR readings in the partograph and perform other management as indicated.

### 2.9. Outcomes

The primary outcome measure was FHR defined as normal (120 to160 beats/min throughout labour and delivery) or abnormal (absent, <120 or >160 beats/min lasting for at least two min) in the continuous Moyo despite repositioning of the Moyo sensor, and with three abnormal assessments at different sites in the intermittent Doppler arm.

Secondary outcomes included the Apgar score at one and five minutes (abnormal was defined as an Apgar score <7); mode of delivery (vaginal delivery, CS, assisted breech, and vacuum extraction); perinatal outcome at birth (i.e., normal, admission to the neonatal unit, or FSB), outcome at 24-h (i.e., normal, still admitted to the neonatal unit, or END); and composite perinatal outcomes at birth and 24-h (normal, admission in neonatal unit, FSB, and END). Apgar score <7 at five minutes was used as a surrogate measure of birth asphyxia [25]. Mode of delivery was dichotomized into two categories (i.e., vaginal, including vacuum delivery, and CS) due to relatively fewer cases in the vacuum delivery category. Time intervals included admission to abnormal FHR detection, admission to delivery, from abnormal FHR detection to delivery, and last FHR assessment to delivery. After detection of abnormal FHR, recorded intrauterine resuscitation included discontinuing oxytocin, changing maternal position, administering intravenous fluids, and provision of oxygen.

### 2.10. Trial Monitoring and Stopping Rules

The trial was monitored by an independent data monitoring committee comprising one statistician and one paediatrician aimed at protecting participant exposure to unreasonable risks. Discontinuation was planned in case of imbalances in serious adverse effects (FSB and END). Blinded data analysis was conducted mid-way through the trial and the committee recommended continuation of the study.

### 2.11. Sample Size Estimation

Historical data showed that when using the hand-held Doppler, abnormal FHR was detected in 4.5% of low-risk deliveries. We postulated that continuous assessment of FHR using Moyo would detect a minimum of 7% of abnormal FHR. To detect these differences at a significance level of 0.05 with 80% power, a minimum of 1350 cases would be needed in each arm. An additional 10% was added to the sample size to allow for missing data. The final sample size was 2970.

### 2.12. Data Management

Data collection was conducted by trained research nurses (at least 2 per shift) filling the CRF. CRFs was cross-checked by the investigators for quality and completeness before entry. All CRFs with queries were returned to the research nurse for verification and correction before data entry. A data entry template was generated in Epi Data by investigators and statistician. All verified data were double-entered by trained data clerks. Then, data was transferred to SPSS for analysis (IBM SPSS Statistics for Windows, Version 23.0, IBM Corp, Armonk, NY, US). Patient information were recorded using confidential codes and kept in a secured place.

### 2.13. Statistical Analysis

Descriptive statistics were expressed as means (standard deviation, SD) or medians (inter quartile range, IQR) for continuous variables and as counts and proportions for categorical variables. Proportions were compared by a Pearson chi-square test. Odds ratios (OR) with respective 95% confidence intervals were calculated as estimates of the effect for categorical variables. Adjusted OR (AOR) using both logistics and multinomial regressions were estimated to account for imbalances in baseline characteristics and for an increase in subject-specific precision. Symmetrically distributed continuous variables were compared by *t*-test, and the Mann-Whitney U was used for skewed data. To adjust for baseline imbalances when comparing skewed time variables, we used linear regression analysis with a natural log-transformed outcome to calculate beta-coefficients. Due to this transformation, we used beta coefficients to estimate the effect size (ES), i.e., relative change in median time in percentages as documented before [26]. A *p*-value < 0.05 was considered significant.

### 2.14. Ethical Clearance

The study was registered at ClinicalTrials.gov Identifier: NCT02790554. All subjects gave their written informed consent for inclusion before they participated in the study. The study was conducted in accordance with the Declaration of Helsinki, and the protocol was approved by both the National Institute of Medical Research in Tanzania (NIMR/HQ/R.8a/Vol. IX/1434) and the Regional Committee for Medical and Health Research Ethics, Western Norway (REK Vest). Local permission was sought from MNH Directorate of Research and Consultancy. Permission to publish was granted by NIMR (NIMR/HQ/P.12 VOL. XXV/57).

## 3. Results

From March 2016 to September 2017, a total of 3547 admitted women were eligible. Of these, 438 were not randomised due to precipitous labour and 136 did not consent to participate in the study. In total, 2973 women were enrolled, 1479 assigned to Moyo and 1494 to Doppler as shown in the study profile (Figure 3).

### 3.1. Maternal, Antenatal, and Perinatal Characteristics

Maternal, antenatal, and perinatal characteristics of the study subjects are shown in Table 1. Maternal mean age was comparable between study arms. The Moyo arm had a lower proportion of preterm deliveries compared to Doppler (12% vs. 17%, *p* ≤ 0.001). Women in the Moyo arm were admitted earlier in labour with a mean cervical dilatation of 4.4 ± 1.5 cm compared to 5.0 ± 1.7 cm in the Doppler arm, *p* ≤ 0.001.

### 3.2. Primary and Secondary Labour and Perinatal Outcomes

Primary and secondary outcomes were adjusted for baseline variables separately. The difference of proportions of preterm births between the two study arms showed a significant influence in the effect measures estimates on most of the perinatal outcomes. Other baseline demographic and clinical characteristics were added in the logistic regression model to increase the precision of subject-specific effect measure estimates (Table 2). There were significantly higher numbers of FHR abnormalities detected in the Moyo versus Doppler arms, i.e., 13.3% versus 9.8%, respectively (AOR = 1.46; 95% CI: 1.16–1.76, *p* = 0.002). There were higher rates of CS in the Moyo as compared to the Doppler arm, i.e., 18.9% versus 12.9%, respectively (AOR = 1.26; 95% CI: 1.01–1.53, *p* = 0.03). AOR of low Apgar scores at one and five minutes did not differ between study arms. AOR of admission to neonatal unit for treatment, FSB, and composite adverse perinatal outcome at delivery were comparable in both study arms after adjustment for gestational age. Similarly, the AOR of admissions to the neonatal unit for treatment, FSB, END, and composite adverse perinatal outcomes at 24-h were not significantly different after adjustment for baseline imbalances.

### 3.3. Comparison of Time Intervals between Continuous Moyo and Intermittent Doppler

Table 3 shows comparisons of linear regression models with natural-log-transformation of skewed time variables between the two study arms. We adjusted for mean admission cervical dilatation since it differed significantly between the two study arms (Table 1). Time from admission to delivery was comparable between study arms (*p* = 0.39). Time interval from admission to abnormal FHR detection was on average 14% shorter in the Moyo as compared to the Doppler arm (*p* = 0.124). Time from last FHR measurement to delivery was on average 12% significantly shorter in the Moyo arm compared to the Doppler arm (*p* = 0.006). Among deliveries with abnormal FHR, the time from detection to delivery was on average 36% significantly longer in Moyo compared to the Doppler arm (*p* = 0.007). Subgroup analysis showed that this difference between the time from detection of abnormal FHR to delivery was 36% significantly longer among vaginal deliveries (*p* = 0.018) and 8% longer among CS deliveries (*p* = 0.680) in Moyo compared to the Doppler arm.

### 3.4. Indications for CS and Intrauterine Resuscitation

Table 4 shows the indications for CS in relation to FHR detection in the two groups. Overall, there was no difference in the proportion of FHR abnormalities in the Moyo compared to Doppler arms (22.9% vs. 17.2%, respectively, *p* = 0.129). There were no differences in FHR abnormalities for the different indications except for obstructed labour group, where FHR abnormalities were detected more often in the Moyo versus the Doppler group (17.3 vs. 7.7%, respectively, *p* = 0.052).

Overall, 85.3% of all foetuses with an abnormal FHR detected received at least one intrauterine resuscitation (87.0% vs. 84.0% for Moyo vs. Doppler, respectively, *p* = 0.281). These interventions included discontinuing oxytocin (38.8% vs. 30.6%, *p* = 0.117), changing maternal position (57.5% vs. 45.5%, *p* = 0.859), and administering intravenous fluids (77.7% vs. 82.9%, *p* = 0.234) for the Moyo versus the Doppler arms, respectively.

### 3.5. Abnormal Foetal Heart Rate Detection, Mode of Delivery, and 24-Hour Perinatal Outcomes in the Continuous Moyo Versus Intermittent Doppler Arms

Figure 4 shows subgroup comparisons of abnormal FHR detection, mode of delivery, and perinatal outcomes between the two arms. Of the 21 perinatal deaths that occurred within 24 h (i.e., 10 FSB and 11 END), 16 were associated with an abnormal FHR detection, equally proportioned in both arms. In cases with abnormal FHR detection, nearly equal proportions of deaths occurred with vaginal deliveries (5.3% vs. 4.4%, *p* = 0.749) in both arms, whereas it was lower in the Moyo compared to the Doppler arm (i.e., 1.6% vs. 9.1%, *p* = 0.077) in CS deliveries, respectively. With a normal FHR, in CS deliveries, there were no deaths in the Moyo arm while there were two deaths in the Doppler arm. For vaginal deliveries, two of the three deaths in the Moyo arm were due to congenital malformation and one due to foetal distress.

## 4. Discussion

This is the first randomised controlled study comparing a robust continuous FHR monitoring device (Moyo), developed for LIC settings to intermittent monitoring (Doppler) in an urban resource limited hospital. Use of the Moyo device identified 46% more foetuses with an abnormal FHR compared to Doppler assessments. An abnormal FHR was detected earlier when using the Moyo as compared to Doppler, however, the time from detection to delivery was longer in the Moyo arm. The CS rates were 26% higher in Moyo compared to Doppler although the difference was due to primary causes rather than an abnormal FHR. There were no differences in perinatal outcomes between the two groups after adjustment for baseline imbalances.

The findings from this study are similar to a recent study by our group comparing the Moyo device with a Pinard fetoscope in a rural setting in Tanzania [18]. Thus, there was an increased detection of abnormal FHR and intrauterine resuscitation in the Moyo arm, however, no differences were noted in perinatal outcomes [18]. One potential explanation for the increased and earlier detection of abnormal FHR when using Moyo is likely due to the increased sensitivity and continuous monitoring of the device. Thus, Moyo has an increased detection area, can detect FHR within 5 s, and has a 9-crystal sensor as compared to the single-crystal sensor in the Doppler machine. In addition, the Moyo is equipped with an automatic alarm which beeps in case of sustained abnormal FHR (>3 min), enabling the midwife to record the abnormalities, which are likely missed by intermittent auscultation [17]. Furthermore, the device provides 30-min FHR recording for review, enabling midwives to monitor labour progress accurately, as we recently documented [17]. In addition, we have recently published qualitative assessment among mothers randomised to continuous monitoring with Moyo versus intermittent Doppler assessment, and reported that Moyo was the preferred device [23]. This was due to an interactive maternal-midwife component of Moyo, related to the fact that mothers could continuously hear the foetal heart sounds. This provided reassurance of their babies’ viability [23].

Despite the increased detection of an abnormal FHR, there were no significant differences in perinatal outcomes (Figure 4). There are several potential reasons for this finding. Firstly, the study was performed among relatively low-risk labouring women, who are less likely to have distressed babies, and hence fewer adverse perinatal outcomes. As noted previously, a very large sample would have been needed to detect such small differences in proportions [9,27,28,29], and the study was not powered to do so. Secondly, while an abnormal FHR was detected earlier using continuous rather than intermittent monitoring, there was a significant overall delay to delivery in both the Moyo and Doppler arms, i.e., 73 min vs. 40 min, respectively, potentially leading to more foetal compromise. Recent studies in rural Tanzania have documented adverse perinatal outcomes associated with delayed delivery of babies with detected FHR abnormalities [5,6]. Timely delivery of these babies may have improved perinatal outcomes in both groups. Notably, in this study, the median time from abnormal FHR detection to delivery by CS was as high as 112 and 100 min in the Moyo and Doppler arms, respectively. The recommended time from decision to Caesarean delivery of the distressed baby is less than 30 min as per current international guidelines [30,31]. Potential reasons for this delay may relate to the fact that some of the women scheduled for CS were held back due to other more urgent CS cases [32]. The overall CS rate at MNH is above 50%, and most of these are done on an emergency basis [32]. Additionally, the labour ward and obstetric theatre are situated in two different buildings, hence increasing the time lag from decision to actual CS (Table 3) [21]. Importantly, evidence from high income countries indicates that the use of advanced FHR monitors coupled with timely CS for foetal distress is associated with reduced neonatal hypoxia, seizures, and perinatal deaths [7,27,28].

The higher rates of CS in the Moyo (26%) compared to the Doppler arm is consistent with previous studies and systematic reviews [7,9,28,33]. However, in this study, the higher CS rates were due to primary obstetric causes (such as obstructed labour) rather than the abnormal FHR (Table 4). Furthermore, previous studies have reported that clinicians and midwives may not undertake timely and appropriate interventions once a decision to perform a CS is taken, leading to the foetus being compromised [33,34]. This could have been a challenge in our study as well (with the obstetric theatre located in a different building), especially in the Moyo arm, with higher rates of CS.

### Limitations

In this resource limited setting, the technology to conduct scalp foetal blood gas sampling, and thus, the ability to identify co-existent hypoxia/acidosis, was not available to support the significance of the FHR abnormalities. Moreover, there was an imbalance in the distribution of preterm infants and cervical dilatation on admission between the two randomisation arms; however, these were adjusted for in the regression analysis. Thirdly, due to the nature of the intervention (medical device), it was not possible to blind the health care workers who implemented and assessed the outcomes. In this study, we used simple randomization instead of a randomised block design with different block sizes, which would have minimized any unmasked bias. Fourth, some women were not randomized in this study due to precipitous labour and few were missed due to concurrent multiple admission, which may have made the findings less generalizable. Moreover, this study was designed to detect an abnormal <120 or >160 beats/min or absent FHR. Thus, the degree or persistence of bradycardia or the degree of the FHR variability were not recorded, which may have influenced the outcome. Finally, the study involved low-risk pregnancies with fewer adverse perinatal outcomes than would have been expected in the overall population.

## 5. Conclusions

An abnormal FHR was detected more frequently and earlier when using continuous monitoring with Moyo as compared to intermittent assessments using Doppler. There were no differences in adverse perinatal outcomes; the latter was likely related to the small sample size, a delayed response to delivery, and the low-risk nature of the study population. Studies designed and powered to detect differences in perinatal outcomes among high risk foetuses with timely obstetric responses are recommended [28].

## Figures and Tables

**Figure 1 ijerph-16-00315-f001:**
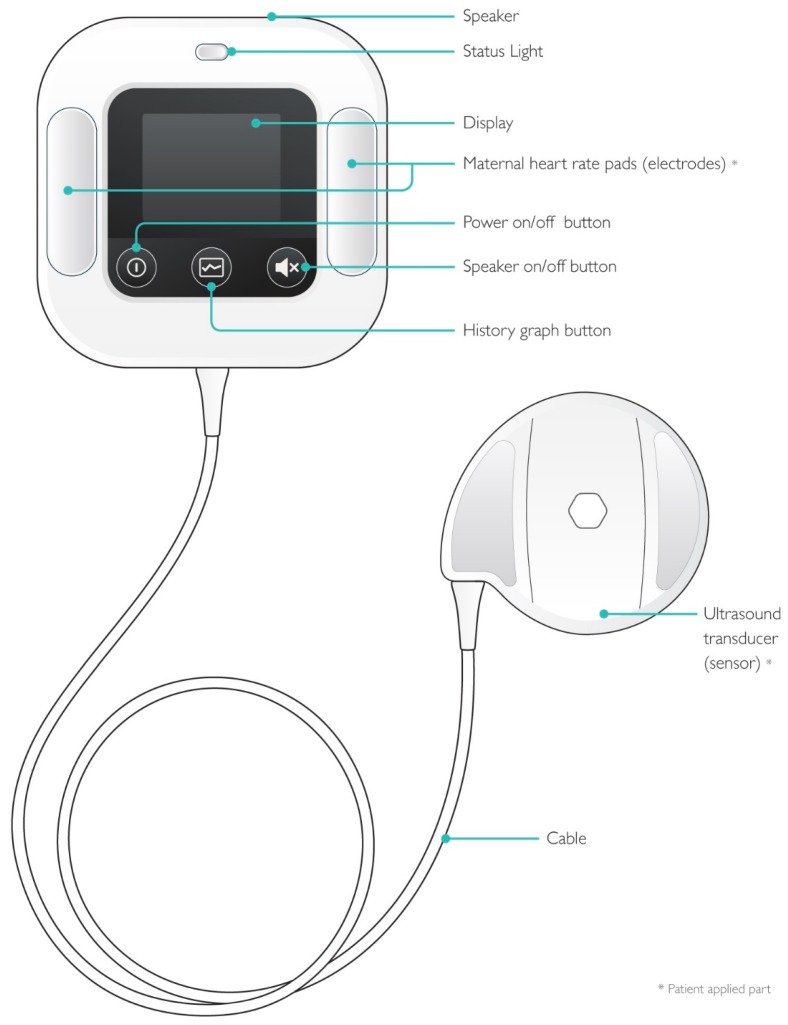
Moyo—the novel continuous FHR (foetal heart rate) monitor (Laerdal Global Health). * patient applied part.

**Figure 2 ijerph-16-00315-f002:**
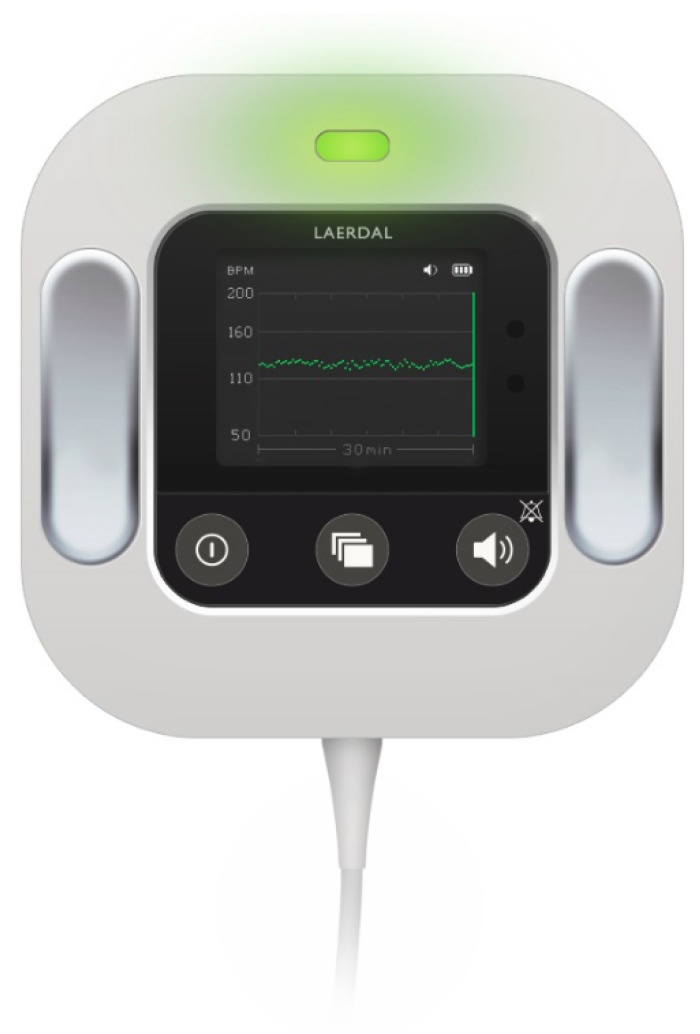
The Moyo FHR monitor with a 30-minutes historical display (Laerdal Global Health).

**Figure 3 ijerph-16-00315-f003:**
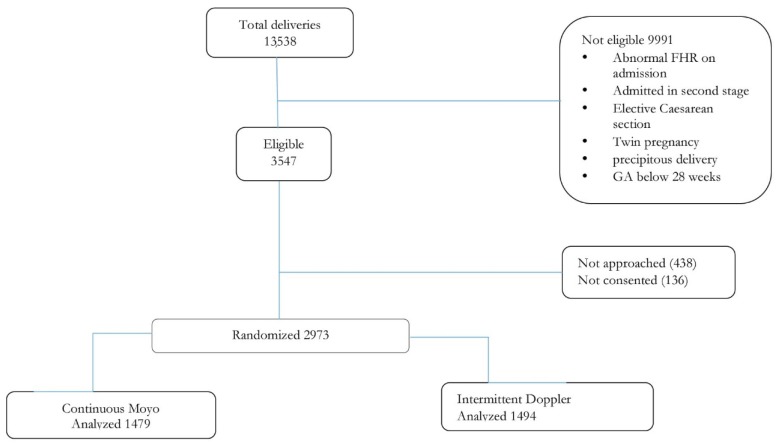
Trial profile.

**Figure 4 ijerph-16-00315-f004:**
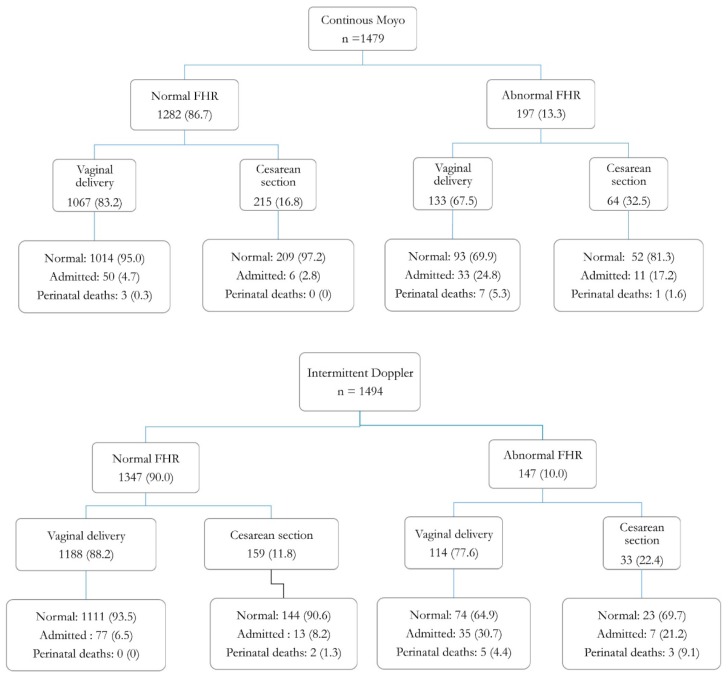
Flow diagrams of foetal heart rate (FHR) detections, mode of delivery, and perinatal outcomes in continuous Moyo and intermittent Doppler.

**Table 1 ijerph-16-00315-t001:** Baseline demographic, clinical, and perinatal characteristics of randomized low risk pregnant women in labour.

Variables	Intermittent Doppler (*n* = 1494)	Continuous Moyo (*n* = 1479)
Age (years)		
Mean (SD)	28.3 (5.6)	27.8 (5.3)
<20	84 (5.6)	66 (4.5)
20–35	1223 (81.9)	1260 (85.2)
>35	187 (12.5)	153 (10.3)
Education		
No/Primary	557 (37.3)	424 (28.7)
Secondary	375 (25.1)	366 (24.7)
College/University	562 (37.6)	689 (46.6)
Marital status		
Married/Cohabiting	1370 (91.7)	1384 (93.6)
Single	124 (8.3)	95 (6.4)
Antenatal care visits		
<4	466 (31.2)	402 (27.2)
≥4	1028 (68.8)	1077 (72.8)
Parity		
Median (IQR)	2 (1, 3)	2 (1, 3)
Prime	576 (38.6)	697 (54.8)
2–4	805 (53.9)	709 (47.9)
>4	113 (7.6)	73 (4.9)
Gestational age (weeks)		
Mean (SD)	37.8 (2.9)	38.1 (2.5)
<37(Preterm)	251 (16.8)	174 (11.8)
≥37(Term)	1243 (83.2)	1305 (88.2)
Birth weight (grams)		
Mean (SD)	2979 (649)	3073 (611)
<2500	273 (18.3)	193 (13.0)
2500–3500	944 (632)	987 (66.7)
>3500	277 (18.5)	299 (20.2)
Cervical dilation on admission (cm)		
Mean (SD)	5.0 (1.7)	4.4 (1.5)
Antenatal problem		
No	1104 (73.9)	1159 (78.4)
Yes	390 (26.1)	320 (21.6)
Obstetric complication		
No	1389 (93.0)	1344 (90.9)
Yes	105 (7.0)	135 (9.1)
Source of admission		
Referred/admitted	623 (41.7)	529 (35.8)
Home	871 (58.3)	950 (64.2)

Data shown as *n* (%) unless otherwise stated. SD: Standard deviation, IQR: Interquartile range.

**Table 2 ijerph-16-00315-t002:** Comparison of labour and perinatal outcomes between intermittent Doppler and continuous Moyo.

Labour and Perinatal Outcomes	Intermittent Doppler (*n* = 1494)	Continuous Moyo (*n* = 1479)	Unadjusted OR * (95% CI)	*p*-Value	AOR (95% CI) **	*p*-Value
FHR during labour						
Normal	1347 (90.2)	1282 (88.4)				
Abnormal	147 (9.8)	197 (13.3)	1.41 (1.12–1.77)	0.003	1.46 (1.16–1.76)	0.002
Mode of delivery						
Vaginal	1302 (87.1)	1200 (81.1)				
CS	192 (12.9)	279 (18.9)	1.58 (1.29–1.93)	0.001	1.26 (1.01–1.53)	0.031
Apgar score at 1st minute						
Normal (≥7)	1361 (91.1)	1373 (92.8)				
Abnormal (<7)	133 (8.9)	106 (7.2)	0.79 (0.61–1.03)	0.082	0.92 (0.69–1.22)	0.555
Apgar score at 5th minute						
Normal (≥7)	1442 (96.2)	1436 (97.1)				
Abnormal (<7)	52 (3.5)	43 (2.9)	0.83 (0.55–1.25)	0.375	0.95 (0.69–1.63)	0.800
Delivery outcomes						
Normal	1338 (89.6)	1361 (92.0)				
Admitted for treatment	147 (9.8)	109 (7.4)	0.73 (0.56–0.94)	0.017	0.88 (0.69–1.17)	0.387
FSB	9 (0.6)	9 (0.6)	0.98 (0.39–2.48)	0.971	1.43 (0.55–1.19)	0.464
Admitted + FSB	156 (10.4)	118 (8.0)	0.74 (0.58–0.96)	0.021	0.91 (0.69–1.19)	0.497
24-h outcome						
Normal	1352 (90.5)	1368 (92.5)				
Still admitted for treatment	132 (8.8)	100 (6.8)	0.75 (0.57–0.98)	0.036	0.93 (0.69–1.24)	0.523
FSB + END	10 (0.7)	11 (0.7)	1.09 (0.46–2.57)	0.849	1.59 (0.65–3.90)	0.345
Admitted + FSB + END	142 (9.5)	111 (7.5)	0.77 (0.59–1.00)	0.051	0.97 (0.72–1.26)	0.706

* Odds ratio for abnormality/poor outcome for Moyo vs. Doppler ** Adjusted for baseline characteristics imbalances. Data shown as *n* (%) unless otherwise stated. OR: Odds ratio, CI: confidence intervals, FHR: Foetal Heart Rate, CS: Caesarean Section, END: Early neonatal deaths; FSB: fresh stillbirths.

**Table 3 ijerph-16-00315-t003:** Comparison of time intervals (in minutes) between intermittent Doppler and continuous Moyo.

Time Intervals	Intermittent Doppler (Median (IQR))	Continuous Moyo (Median (IQR))	Unadjusted β-Coefficient (95% CI)	Unadjusted Effect Size (%)	*p*-Value	Adjusted β-Coefficient (95% CI) *	Adjusted Effect Size (%)	*p*-Value
Admission to abnormal FHR Detection (minute)	*n* = 147	*n* = 197						
	197 (108, 330)	192 (110, 330)	0.00 (−0.19–0.20)	1 (−17–22)	0.962	−0.15 (−0.34–0.04)	−14 (−29–4)	0.124
Admission to delivery	*n* = 1494	*n* = 1479						
	240 (150, 390)	288 (171, 288)	0.14 (0.08–0.19)	15 (8–20)	<0.001	−0.02 (−0.07–0.27)	−2 (−7–31)	0.399
Last FHR to delivery	*n* = 1494	*n* = 1479						
	15 (9, 30)	13 (6, 30)	−0.08 (−0.17–0.01)	−8 (−16–1)	0.082	−0.13 (−0.21–0.04)	−12 (−19–4)	0.006
Abnormal FHR to delivery (All deliveries)	*n* = 147	*n* = 197						
	40 (25, 98)	73 (40, 130)	0.42 (0.20–0.64)	52 (22–90)	0.001	0.31 (0.09–0.53)	36 (9–70)	0.007
Abnormal FHR to delivery (VD)	*n* = 114	*n* = 133						
	30 (20, 52)	54 (30, 94)	0.42 (0.17–0.67)	52 (19–95)	0.001	0.31 (0.05–0.57)	36 (5–77)	0.018
Abnormal FHR to delivery (CS)	*n* = 33	*n* = 64						
	110 (89, 162)	122 (78, 141)	0.08 (−0.23–0.39)	8 (−21–48)	0.496	0.08 (−0.24–0.39)	8 (−21–48)	0.680

* Adjusted for cervical dilatation (by linear regression of natural-log-transformed time intervals); IQR: interquartile range, CI: Confidence Intervals; FHR: foetal heart rate; VD: vaginal delivery, CS: Caesarean Section; All time intervals are in minutes.

**Table 4 ijerph-16-00315-t004:** Comparison of indications for Caesarean section (CS) by foetal heart rate (FHR) abnormalities between Doppler and Moyo.

Indication for CS	Intermittent Doppler *n* = 192		Continuous Moyo *n* = 279		*p*-Value
	Normal FHR *n* = 159 (82.8)	Abnormal FHR *n* = 33 (17.2)	Normal FHR *n* = 215 (77.1)	Abnormal FHR *n* = 64 (22.9)	
Obstructed labour	72 (92.3)	6 (7.7)	100 (82.6)	21 (17.3)	0.052
Persistently abnormal FHR	0 (0)	21 (100)	0 (0)	39 (100)	NA
Prolonged labour	53 (100)	0 (0)	85 (98.8)	1 (1.2)	NA
Others	34 (85.0)	6 (15.0)	30 (90.9)	3 (9.1)	0.584

Data is shown as *n* (%), NA: Not applicable because one of the cells contains a zero value.

## Data Availability

Raw data is available only on request. One can contact the Head of Research, teaching and Consultancy unit of Muhimbili National Hospital: Address, P.O. Box 65000. City, Dar es Salaam. Phone, +25-5222151599. Fax, +25-5222150534. Email: info@mnh.or.tz. No additional data available.

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
