# Peer review of "Effectiveness of a Novel Continuous Doppler (Moyo) Versus Intermittent Doppler in Intrapartum Detection of Abnormal Foetal Heart Rate: A Randomised Controlled Study in Tanzania"

_ijerph, 2019, doi:10.3390/ijerph16030315_

Round 1
Reviewer 1 Report
Overall, the manuscript by Kamala et al. is well written. Results from previous studies of the same group already showed the advantages of Moyo. The current study further compared two abnormal FHR detection systems (i.e., Moyo vs Doppler). The authors gave a very clear introduction and detailed experimental design. Materials and methods are thorough. The results are properly presented and the discussion is adequate. They also discussed the limitations of the study. The only concern is that several figures are in low resolution. I suggest the authors should replace Figure 1a, Figure 2, and Figure 3 with high resolution figures. After the minor revision, the manuscript should be ready for publication at IJERPH.
Author Response
Point 1: The only concern is that several figures are in low resolution. I suggest the authors should replace Figure 1a, Figure 2, and Figure 3 with high resolution figures.
Response 1: We appreciate this comment. We have submitted revised and updated figures with high resolution in as separate files
Reviewer 2 Report
1. Line 39-40: It is not clear whether this data is global/local/some specific country.
2. Line 54-55: If it is already proved in the previous RCTs by the same authors then What is
new with this study?
3. Line 113: Unmasked RCTs are prone to a number of biasses. How did authors manage that?
4. Line 208: 438 women were not approached for randomization. Why? it itself is a big source
of potential bias which makes the results of this study completely reliable. Authors are in
need to give a valid reason for this.
Author Response
Thank you very much for your comments, please see the responses in the attachment.

Reviewer 3 Report
The presented study is written correctly in terms of methodology, however the topic is not innovative and the conclusions do not bring anything new to the current state of knowledge. In the introduction, the authors do not refer to the latest publications, eg item No. 13: in 2009 an update appeared and should be quoted. Please refer to the latest research. In table 1 and 4, the significance (p) results should be added. Figure 2 is out of focus, difficult to read - please correct In the discussion, the authors discuss in most of the results of their earlier work, there is no reference to the latest research published in the topic of intrapartum detection of foetal heart rate. The discussion did not cover all the outcomes described in the study, eg regarding perinatal death - please complete.Author Response
Thank you very much for your comments, please see the attachment in this message.

Round 2
Reviewer 2 Report
Point 3:
Again, this is an unmasked RCT and is prone to bias. In their discussion section, the authors need to acknowledge that they did not us a randomised block design with randomly chosen block sizes. Such a design would have minimize unmasked bias. (Ref: Efird J. Blocked randomization with randomly selected block sizes. Int J Environ Res Public Health. 2011;8(1):15-20)
Point 4:
Additionally, I would suggest that the authors simply mention in their limitation section that 438 women were not approached for randomization and it may resulted in selection bias.
Author Response
Response to Reviewer 2 Comments
Point 3: Again, this is an unmasked RCT and is prone to bias. In their discussion section, the authors need to acknowledge that they did not use a randomised block design with randomly chosen block sizes. Such a design would have minimized unmasked bias. (Ref: Efird J. Blocked randomization with randomly selected block sizes. Int J Environ Res Public Health. 2011;8(1):15-20)
Response 3: We appreciate this comment. We have discussed this limitation on page 14 as follows:
“In this study we used simple randomization instead of randomised block design with different block sizes which would have minimize unmasked bias”
Point 4: Additionally, I would suggest that the authors simply mention in their limitation section that 438 women were not approached for randomization and it may result in selection bias.
Response 1: We appreciate this comment. However, we believe that IF the midwives were to have a bias towards approaching the women, the results would be less generalizable, but not biased (since it happens before randomization). We have mentioned this limitation on page 14 and 15 as follows:
“Some women were not randomized in this study due to precipitous labour and few missed due concurrent multiple admission which may have made the findings less generalizable”
Reviewer 3 Report
Accept in present form.
Author Response
Response to Reviewer 3 Comments
Comment from reviewer Accept in present form
Response: Thank you for accepting our revision